# Short-Term Effects of Cover Grass on Soil Microbial Communities in an Apple Orchard on the Loess Plateau

**Pan Wan * and Ruirui He**

College of Forestry, Northwest A&F University, Yangling 712100, China; hrr19911223@163.com

* Correspondence: wp7413841@nwsuaf.edu.cn

**Abstract:** Grass cover may improve soil environmental conditions in apple orchards. However, the mechanisms for how the soil microbial community changes after cover grass treatments are not well understood. In this study, we analyzed soil properties, microbial community diversity and composition in an apple orchard after being covered with native wild grasses for 3 years on the Loess Plateau, China. The ratios of cover grass were 0% (no cover, NC), 20% (low-intensity cover, LIC), 40% (moderate-intensity cover, MIC$_1$), 60% (moderate-intensity cover, MIC$_2$) and 80% (high-intensity cover, HIC). Meanwhile, the relationships between soil nutrients, cover grass properties, and microbial communities was analyzed by redundancy analysis and Pearson correlations. The results showed that cover grass altered the bacterial community composition, and significant changes at the phylum level were mainly caused by Proteobacteria, Bacteroidetes and Chloroflexi. Compared with NC, the abundance of Proteobacteria was lower in LIC, and the abundance of Bacteroidetes was lower in LIC, MIC$_1$ and MIC$_2$, while that of Chloroflexi was higher in LIC. LIC and MIC$_1$ were the only cover grass intensities that altered the soil fungal community composition; there were no significant differences at the phylum level. The changes in the soil microbial community at the given phyla may be related to the change in soil available nitrogen content caused by cover grass. Here, we demonstrate that cover grass changed the soil microbial community, and the changes may be attributed to the given phyla in the bacterial community; soil copiotrophic groups (e.g., Proteobacteria and Bacteroidetes) were found to be at lower abundance in the low-intensity cover grass.

**Keywords:** cover grass; soil properties; soil microorganisms; apple orchard

## 1. Introduction

Soil microbial communities are affected by aboveground plants because plant type can directly affect the plant residues, root systems and root exudates that, in turn, alter soil properties and nutrients [1–3]. Hence, in agricultural ecosystems, the soil microbial community composition strongly depends on the aboveground crop types [4,5]. Furthermore, agricultural soils are frequently affected by agricultural practices, such as ploughing, covering and fertilizing, which may more easily indirectly influence the soil microbiota [6–8]. Thus, exploring changes in microbial communities after agricultural production is an indispensable component of understanding ecological processes in soils.

Grass cover is an effective practice in orchard soil management [9–11], which can reduce the soil bulk density [12,13], increase organic carbon and nutrient contents [11,14–16] and improve soil enzyme and microbial activities [17,18]. Accordingly, researchers have shown that grass cover influences soil microbes because of changes in edaphic properties [6,19]. Despite some research on effects of groundcover management on the soil microbial community, most of these studies are focused on the cover method and plant species, while little attention has been focused on how cover grass and its intensities affect the soil microorganisms and relationships within the taxa in the microbial community populations.

The apple industry has been a pillar of the Chinese agricultural system [20,21]. Planting apple trees promotes the national policy of returning farmland to forest and grassland

and further contributes to improvements in the environment [22–24]. In addition, the income provided by the apple industry has lifted several people out of poverty, improved the rural economy and promoted the long-term development of agricultural industrialization [25,26]. Grass cover could reduce soil and water losses, thereby improving the fertilizer utilization efficiency and soil conditions in apple orchards [11,16]. Grass cover treatment has been shown to impact soil temperature and water content, soil microbial carbon, apple tree growth and apple yields in apple orchards [11,25,26].

To test whether the soil microbial communities respond to the grass cover, we compared the diversity, composition and networks of soil microbial communities in an apple orchard with different intensities of cover grass for 3 years in the Loess Plateau. We hypothesized that grass cover altered the soil microbial community compositions over the short-term, and these changes vary at different grass cover intensities; additionally, the changes in the soil microbial community may be related to the change in soil properties caused by cover grass. Hence, the objectives of this study were to: (1) analyze and compare the changes in soil and cover grass properties, microbial community compositions, and diversity, and (2) explore the effects of soil and cover grass properties on the soil microbial community. The results of this study will contribute to our understanding of the fundamental soil ecological processes in apple orchards following coverage with grasses.

## 2. Materials and Methods

### 2.1. Research Area

The research zone was located in Changwu field experimental station of Northwest A&F University (east longitude 107°40′, north latitude 35°13′, altitude of 1220 m), northwest of Shaanxi Province, Loess plateau, China. The area has a temperate continental monsoon climate, warming slowly in spring, hot in summer, cool in autumn and cold in winter. The annual average temperature and precipitation is 9.4 °C and 560 mm, respectively. The annual dryness is 1.38, the annual sunshine duration is 2230 h, the annual total radiation is 484 kJ cm$^{-2}$ and the frost-free period is 171 days [27]. The soil type is a uniform loam of loess deposits with organic carbon of 6.50g/kg, soil total nitrogen 0.62 g/kg, available nitrogen 37.0 mg/kg, available phosphorus 3.0 mg/kg and available potassium 129.3 mg/kg [28].

### 2.2. Experimental Design

#### 2.2.1. Grass Cover Intensity

Five Fuji apple (*Malus pumila* Mill.) trees (full productive age, 15 years) were selected from an orchard for this experiment; the apple orchard was converted from cropland in 2003. The planting interval for the apple trees was 2 m × 4 m. The apple orchards in this area were not irrigated, and rainfall was the only source of water. The soil underneath the trees was kept weed-free using tillage. Apple trees were pruned in April and the fruits were harvested in October every year. Litter in the orchard was removed in the autumn.

In the spring of 2017, five apple fields were randomly selected, with each containing 5 plots of the same size (8 m × 8 m) (each plot contained 8 apple trees). Each plot was covered with grass at different cover ratios. The ratios of cover grass were no cover (NC, 0% cover grass), low-intensity cover (LIC, 20% cover grass), moderate-intensity cover (MIC$_1$, 40% cover grass and MIC$_2$, 60% cover grass) and high-intensity cover (HIC, 80% cover grass). In total, 25 plots were established, each cover grass treatment was 5 replicates. The cover grass consisted of native wild grass species, primarily *Stellaria media* (L.) Cyr. When the grass had grown to 40–50 cm it was mown to approximately 15 cm in the cover grass area (4 to 6 times a year). The grass outside of the experimental plots was completely weeded once every 7 to 10 days.

#### 2.2.2. Soil Sampling and Grass Collection

In the autumn of 2020, soil samples were separately taken from 25 plots subjected to five cover grass intensity treatments. In each sampling plot, five soil samples with an "S" shaped distribution were randomly collected from topsoil (0–30 cm) and were mixed

together. Thus, in total, 25 composite samples were collected, namely, each treatment was 5 replicates soil samples. The collected soil samples were stored at low temperatures and brought back to the laboratory as soon as possible. Each sample was divided into two parts for analysis of soil microbial community and soil chemical.

In addition, five 1 m × 1 m plots were investigated in each plot to study the cover grass properties (Table 1). Hence, 125 grass samples were collected, and the mean value of five grass samples in each quadrat represents the plot. The grasses were harvested from each quadrat separately and gently washed with deionized water to analyze indicators, such as the biomass, by drying at −80 °C to a constant mass, and then weighed [29].

**Table 1.** Soil and cover grass properties for different treatments.

| Properties | No Cover (0%) | Low-Intensity Cover (20%) | Moderate-Intensity Cover (40%) | Moderate-Intensity Cover (60%) | High-Intensity Cover (80%) | *F* | *p*-Value |
|---|---|---|---|---|---|---|---|
| Soil physical properties | | | | | | | |
| Soil temperature/°C | 14.85 ± 1.63 a | 15.18 ± 1.47 a | 15.03 ± 1.40 a | 15.46 ± 1.82 a | 16.09 ± 1.98 a | 0.418 | 0.794 |
| Soil water content/% | 20.32 ± 1.51 a | 19.73 ± 0.72 a | 20.73 ± 3.20 a | 19.80 ± 1.79 a | 19.07 ± 0.67 a | 0.589 | 0.674 |
| Soil chemical properties | | | | | | | |
| pH | 7.86 ± 0.13 a | 8.12 ± 0.09 a | 8.00 ± 0.18 a | 8.05 ± 0.19 a | 8.11 ± 0.16 a | 2.363 | 0.088 |
| SOC (g·kg$^{-1}$) | 11.84 ± 0.73 a | 11.59 ± 0.81 a | 10.79 ± 0.64 a | 10.83 ± 0.94 a | 11.39 ± 0.99 a | 1.555 | 0.225 |
| TN (g·kg$^{-1}$) | 1.14 ± 0.10 a | 0.98 ± 0.07 b | 0.97 ± 0.09 b | 0.94 ± 0.08 b | 0.93 ± 0.11 b | 4.377 | 0.011 |
| AN (mg·kg$^{-1}$) | 54.04 ± 14.96 a | 40.32 ± 4.24 b | 41.16 ± 7.11 b | 36.96 ± 2.34 b | 35.00 ± 4.08 b | 4.400 | 0.010 |
| TP (g·kg$^{-1}$) | 1.25 ± 0.16 a | 1.23 ± 0.07 a | 1.21 ± 0.09 a | 1.13 ± 0.10 a | 1.26 ± 0.24 a | 0.629 | 0.647 |
| AP (mg·kg$^{-1}$) | 49.08 ± 21.96 a | 37.78 ± 11.87 a | 34.12 ± 5.47 a | 28.36 ± 15.64 a | 37.10 ± 9.81 a | 1.439 | 0.258 |
| TK (g·kg$^{-1}$) | 21.60 ± 0.19 a | 21.60 ± 0.20 a | 21.59 ± 0.31 a | 21.14 ± 0.57 a | 21.18 ± 0.21 a | 2.660 | 0.063 |
| AK (mg·kg$^{-1}$) | 499.40 ± 301.72 a | 392.40 ± 67.45 a | 368.60 ± 72.38 a | 373.60 ± 182.91 a | 323.20 ± 59.45 a | 0.805 | 0.557 |
| MBC (mg·kg$^{-1}$) | 37.99 ± 13.97 a | 39.13 ± 19.41 a | 44.38 ± 16.76 a | 32.86 ± 6.16 a | 40.46 ± 14.02 a | 0.402 | 0.805 |
| MBN (mg·kg$^{-1}$) | 6.17 ± 1.23 a | 9.38 ± 6.06 a | 8.60 ± 3.65 a | 6.68 ± 2.58 a | 9.40 ± 0.88 a | 0.985 | 0.438 |
| Cover grass | | | | | | | |
| GC (g·kg$^{-1}$) | – | 421.01 ± 19.81 a | 421.43 ± 18.57 a | 408.19 ± 37.52 a | 415.25 ± 25.1 a | 0.276 | 0.842 |
| GN (g·kg$^{-1}$) | – | 17.23 ± 3.04 a | 14.65 ± 2.51 a | 15.83 ± 1.20 a | 17.26 ± 3.27 a | 1.046 | 0.399 |
| GP (g·kg$^{-1}$) | – | 2.96 ± 0.36 a | 2.36 ± 0.39 a | 2.43 ± 0.30 a | 2.78 ± 0.49 a | 2.651 | 0.084 |

The values in the table are mean values (± SD, *n* = 5). Significant differences among different vegetation types are indicated by different letters at the 0.05 level. T indicates temperature, SWC indicates soil water content, SOC indicates soil organic carbon, AN indicates available nitrogen, TN indicates total nitrogen, TP indicates total phosphorus, AP indicates available phosphorus, AK indicates available potassium, TK indicates total potassium, MBC indicates microbial biomass carbon, MBC indicates microbial biomass nitrogen, GB indicates cover grass biomass, GC indicates cover grass carbon, GN indicates cover grass nitrogen, and GP indicates cover grass phosphorus.

### 2.2.3. Soil Microbial Community and Soil Properties

The soil microbial community was investigated by high-throughput sequencing of the highly variable regions of 16S rRNA and 18S rRNA. The main steps of the experiment included the extraction of total soil DNA, PCR amplification, and Illumina MiSeq sequencing. Briefly, the details of the experiment are as follows: (1) Total DNA from the soil was extracted using the E.Z.N.A.® Soil DNA Kit (Omega Bio-Tek, Norcross, GA, USA), and microbial DNA was extracted from each sample (25 soil samples). 0.25 g of soil used for DNA extraction and 5 replicates for each analysis in our study. One percent agarose gel electrophoresis was used to check the integrity of DNA, and a NanoDrop2000 (Thermo Scientific, Wilmington, NC, USA) was used to check the purity and concentration of DNA. (2) For bacteria, 515F (5′-GTGCCAGCMGCCGCGG-3′) and 907R (5′-CCGTCAATTCMTTTRAGTTT-3′) primers were used for PCR amplification of the V3–V5 variable region. For fungi, SSU0817F (5′-TTAGCATGGAATAATRRAATAGGA-3) and 1196R (5′-TTAGCATGGAATAATRRAATAGGA-3′) and 1196R (5′-CCGTCAATTCMTTTRAGTTT-3′) primers were used. For PCR, a 20 μL reaction included the 5 × FastPfu buffer and FastPfu buffer (4 μL each), 2.5 mM dNTPs (2 μL), upstream and downstream primers (each 0.8 μL), DNA template (10 ng), and ddH$_2$O to total 20 μL. PCR amplification conditions were as follows: 95 °C for 3 min for predenaturation, 9 cycles (95 °C for 30 s, 55 °C for 30 s, 72 °C for 45 s), and finally 72 °C for 10 min for extension. (3) According to the standard protocol of Majorbio Bio-Pharm Technology Co., Ltd., (Shanghai, China), on the Illumina MiSeq platform (Illumina, San Diego, CA, USA), the purified amplicons were pooled

equimolar and subjected to paired-end sequencing (2 × 250 bp). For the more detailed method, see the reference [30].

Soil water content (SWC) was determined by drying method [31] and soil temperature (T) was measured using the soil temperature instrument (Table 1). Soil pH, organic carbon (SOC), total nitrogen (TN), available nitrogen (AN), total phosphorus (TP), available phosphorus (AP), total potassium (TK) and available potassium (AK) were measured; for details of the measurement method, refer to Wan et al. [32]. The concentrations of soil microbial biomass carbon (MBC) and nitrogen (MBN) were analyzed by fumigation extraction [33].

### 2.2.4. Cover Grass Chemical Analysis

Dry grass samples were obtained to determine the content of C, N and P. Total C and N concentrations were determined by an elemental analyzer and total P concentration of dry samples was determined by the molybdenum blue colorimetric method [27].

### *2.3. Statistical Analysis of Data*

Fastp software was used for quality control of the original sequences, FLASH software was used for splicing, and UCHIME software was used to eliminate chimaeras and obtain optimized sequences (version 4.2, http://drive5.com/uchime/ accessed on 18 March 2021, San Francisco, CA, USA). UPARSE software (http://drive5.com/uparse/ accessed on 18 March 2021, version 7.1) was used to analyze 97% OTUs by sequence similarity clustering [34]. The taxonomy of each 16S rRNA and 18S rRNA gene sequence was analyzed using the RDP Classifier algorithm (http://rdp.cme.msu.edu/accessed on 18 March 2021) against the Silva (SSU123) database using a confidence threshold of 70%. The sequencing depth have been rarefied to the same sequence number per sample. According to OTU abundance information, Mothur (1.30.2) software was used to perform diversity analysis on samples, including the ACE index, Chao index, and Shannon index. The composition of microbial communities was counted at the phylum classification level. One-way analysis of variance (ANOVA) was used to estimate variations in soil properties (temperature, water content, pH, SOC, TN, AN, TP, AP, TK, AK, MBC and MBN), cover grass characteristics (biomass, GC, GN and GP), microbial community diversity and richness indices (observed species, Chao index, Shannon index, ACE index and number of OTUs). The partial least squares discriminant analysis (PLS-DA) was performed on OTU data to discriminate the microbial community profiles of different treatments [35]. Redundancy analysis (RDA) and Pearson correlations were used to analyze the relationships between soil nutrients, cover grass, and microbial (bacterial and fungal) communities [36]. Pearson correlation analysis was used to assess the relationships between dominant microbial communities and environmental variables. All data were analyzed on the free online platform of the Majorbio I-Sanger Cloud Platform (http://www.i-sanger.com accessed on 20 March 2021) (Shanghai Majorbio Bio-pharm Technology Co., Ltd., Shanghai, China).

## 3. Results

### *3.1. Cover Grass and Soil Properties*

The cover grass decreased the available nitrogen (AN) and total nitrogen (TN) contents ($p < 0.05$); however, the soil total N and available N displayed no significant difference among the different cover grass treatments over the short term ($p > 0.05$). Other soil nutrients and cover grass properties were not significantly influenced by the cover grass ($p > 0.05$) (Table 1).

### *3.2. Microbial Communities*

The average number of bacterial sequences was 58,196 with a mean length of 418 bp, and the average number of fungal sequences was 96,114 with a mean length of 242 bp (Tables S1 and S2). Bacterial sequences were clustered into 1768 OTUs and the fungal sequences were clustered into 622 OTUs using the Bayesian classifier at a 97% similarity

level. The rarefaction curves of the 25 soil samples flattened, indicating that the detection results reflected the bacterial or fungal community of the soil samples (Figure S1a,b). Analysis of OTUs determination at the 97% similarity level showed that the soil diversity and richness, the observed taxa, Chao index, Shannon index, ACE index and the number of OTUs for bacteria and fungi displayed no significant difference among the different treatments ($p > 0.05$) (Table 2).

**Table 2.** Soil diversity and richness for different treatments.

| Indicators | No Cover (0%) | Low-Intensity Cover (20%) | Moderate-Intensity Cover (40%) | Moderate-Intensity Cover (60%) | High-Intensity Cover (80%) | *F* | *p*-Value |
|---|---|---|---|---|---|---|---|
| Bacterial | | | | | | | |
| Observed species | 419 ± 4 a | 417 ± 9 a | 419 ± 5 a | 414 ± 6 a | 414 ± 10 a | 0.562 | 0.693 |
| Chao index | 1728.18 ± 10.32 a | 1746.13 ± 8.83 a | 1731.33 ± 25.37 a | 1720.29 ± 9.84 a | 1739.18 ± 14.34 a | 2.199 | 0.106 |
| Shannon index | 6.24 ± 0.13 a | 6.43 ± 0.09 a | 6.30 ± 0.28 a | 6.24 ± 0.15 a | 6.39 ± 0.09 a | 1.440 | 0.257 |
| ACE index | 1718.95 ± 11.21 a | 1731.13 ± 14.05 a | 1723.38 ± 22.62 a | 1710.08 ± 7.99 a | 1729.71 ± 13.23 a | 1.711 | 0.187 |
| Number of OTUs | 1651 ± 6 a | 1694 ± 23 a | 1678 ± 13 a | 1661 ± 10 a | 1671 ± 54 a | 1.777 | 0.173 |
| Fungi | | | | | | | |
| Observed species | 46 ± 5 a | 45 ± 7 a | 47 ± 9 a | 48 ± 4 a | 45 ± 6 a | 0.192 | 0.939 |
| Chao index | 307.34.08 ± 41.84 a | 286.94 ± 67.80 a | 310.10 ± 63.43 a | 308.69 ± 38.02 a | 312.87.10 ± 59.25 a | 0.177 | 0.948 |
| Shannon index | 3.46 ± 0.28 a | 3.89 ± 0.37 a | 3.62 ± 0.36 a | 3.53 ± 0.47 a | 3.37 ± 0.27 a | 1.520 | 0.234 |
| ACE index | 312.29 + 35.84 a | 302.51 + 60.43 a | 345.37 + 25.17 a | 311.62 + 40.00 a | 315.24 + 58.33 a | 0.631 | 0.646 |
| Number of OTUs | 284 ± 51.40 a | 265 ± 62 a | 264 ± 80a | 271 ± 53 a | 279 ± 59 a | 0.096 | 0.983 |

The values in the table are mean values (±SD, *n* = 5). Significant differences among different vegetation types are indicated by different letters at the 0.05 level.

Proteobacteria, Acidobacteria, Actinobacteria, Chloroflexi, Gemmatimonadetes, Bacteroidetes, Planctomycetes, Rokubacteria, Nitrospirae, and Verrucomicrobia were the dominant bacterial phyla (Figure 1a). For the bacterial community (Table S3), the LIC treatment had a lower Proteobacteria abundance than the NC ($p < 0.05$) but had a higher Chloroflexi abundance than the NC and the other treatments ($p < 0.05$). The abundance of Gemmatimonadetes in the MIC$_2$ and HIC was significantly lower than that in the NC ($p < 0.05$). In addition, the abundance of Bacteroidetes in the LIC, MIC$_1$ and MIC$_2$ was obviously lower than that in the NC ($p < 0.05$), and the Nitrospirae abundance in the HIC was higher than that in the NC ($p < 0.05$). The dominant bacterial orders are shown in Figure S2a. Sphingomonadales, Betaproteobacteriales, Gemmatimonadales, Rhizobiales, Xanthomonadales, Micrococcales, Myxococcales and Cytophagales were the most abundant orders. The Sphingomonadales abundance in the LIC, MIC$_1$, MIC$_2$ and HIC was lower than that in the NC ($p < 0.05$) (Table S4).

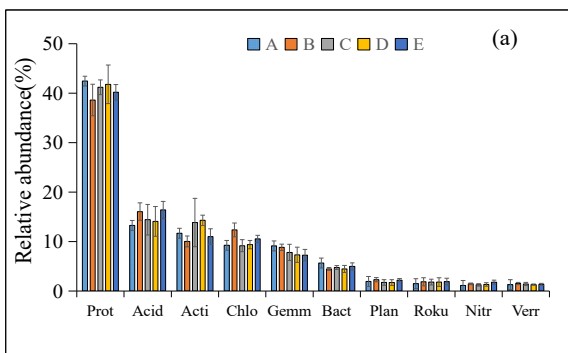
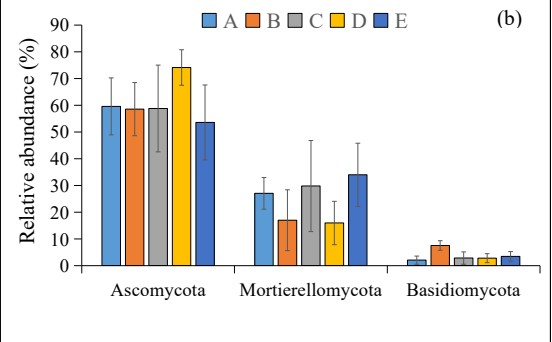

**Figure 1.** Relative abundance of the dominant groups of bacterial (**a**) and fungal (**b**) communities at the phylum level among the different treatments. A indicates 0% (NC), B indicates 20% (LIC), C indicates 40% (MIC$_1$), D indicates 60% (MIC$_2$) and E indicates 80% (HIC). Prot indicates Proteobacteria, Acid indicates Acidobacteria, Acti indicates Actinobacteria, Chlo indicates Chloroflexi, Gemm indicates Gemmatimonadetes, Bact indicates Bacteroidetes, Plan indicates Planctomycetes, Roku indicates Rokubacteria, Nitr indicates Nitrospirae and Verr indicates Verrucomicrobia.

The Ascomycota abundance in the HIC was significantly lower than in the MIC$_2$ ($p < 0.05$), and the relative abundance of Mortierellomycota in the HIC was significantly higher than that in the LIC and MIC$_2$ ($p < 0.05$) (Table S5). The dominant Ascomycetes classes observed here were Sordariomycetes, Dothideomycetes, and Leotiomycetes. Mortierellomycetes, Agaricomycetes, and Tremellomycetes were the most abundant classes observed within the Mortierellomycota and Basidiomycete phyla (Figure S2b). The relative abundance of Mortierellomycetes in the HIC was significantly higher than that in the LIC and MIC$_2$ ($p < 0.05$) (Table S6).

Partial least squares discriminant analysis (PLS-DA) results showed that the soil bacterial communities in the NC, MIC$_1$, MIC$_2$ and HIC treatments tended to fall into two groups, while the LIC measure was obviously separated from the NC and other measures (Figure 2a). Similarly, the fungal communities in the NC, MIC$_2$ and HIC measures tended to group together, while the NC measure was obviously separated from the LIC and MIC$_1$ measures, which were clearly separated from one another (Figure 2b).

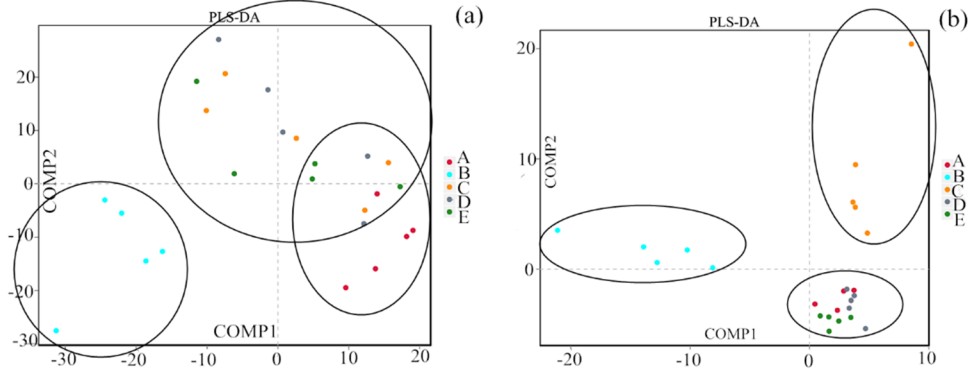

**Figure 2.** Partial least squares discriminant analysis (PLS-DA) of the bacterial (**a**) and fungal (**b**) community compositions among the different treatments. A indicates 0% (NC), B indicates 20% (LIC), C indicates 40% (MIC$_1$), D indicates 60% (MIC$_2$) and E indicates 80% (HIC).

### 3.3. Relationships between Cover Grass, Soil Properties and Microbial Communities

The RDA showed that the compositions of soil bacterial and fungal communities explained 78.10% and 53.98% of the total environmental variations, respectively (Figure 3). The soil properties, i.e., pH, TN, AN, AP, AK and SOC were related to the bacterial communities, while TP, TK, AP, AK, SOC and MBN were related to the fungal communities. Moreover, the RDA results showed that the carbon and phosphorus contents of the cover grass had significant impacts on the bacterial communities (Table 3).

Gemmatimonadetes, Bacteroidetes and Proteobacteria abundances were markedly positively correlated with SOC, TK, AP, AN and AK. Actinobacteria abundance was obviously positively correlated with TK, AN and AK. Chloroflexi abundance was clearly positively correlated with AP and AK. Proteobacteria, Bacteroidetes, Actinobacteria, Gemmatimonadetes, Verrucomicrobia, Acidobacteria, Chloroflexi and Nitrospirae abundances were negatively correlated with GP and GB, and Bacteroidetes abundance was also negatively correlated with GC and GN (Figure 4a). Additionally, Ascomycota and Mortierellomycota abundances were markedly positively correlated with SOC and negatively correlated with GP (Figure 4b).

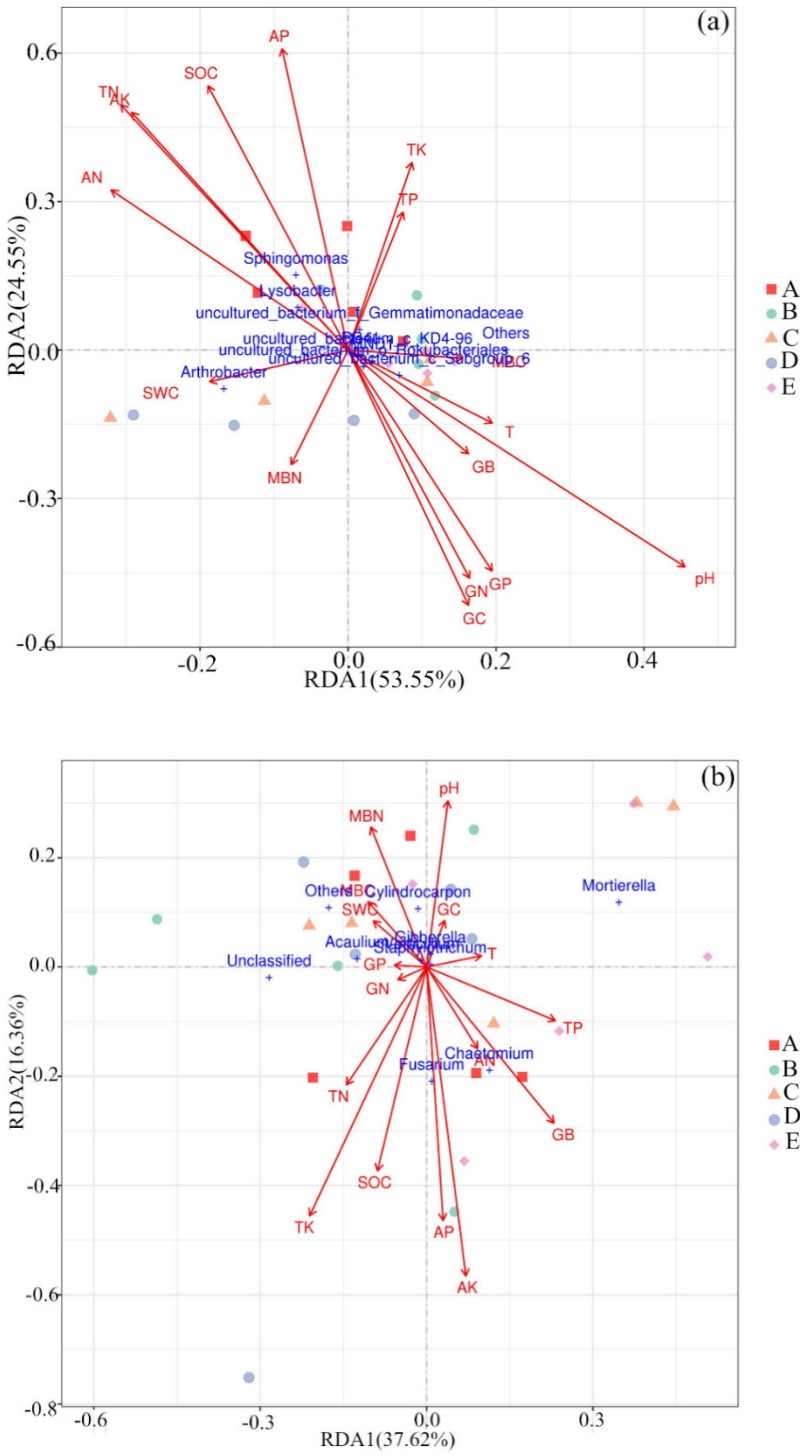

**Figure 3.** Redundancy analysis (RDA) of the relationships between soil and cover grass properties and microbial communities. T indicates soil temperature, SWC indicates soil water content, SOC indicates soil organic carbon, AN indicates available nitrogen, TN indicates total nitrogen, TP indicates total phosphorus, AP indicates available phosphorus, AK indicates available potassium, TK indicates total potassium, MBC indicates microbial biomass carbon, MBC indicates microbial biomass nitrogen, GB indicates cover grass biomass, GC indicates cover grass carbon, GN indicates cover grass nitrogen, and GP indicates cover grass phosphorus A indicates 0% (NC), B indicates 20% (LIC), C indicates 40% ($MIC_1$), D indicates 60% ($MIC_2$) and E indicates 80% (HIC). (**a**) indicates the bacterial community and (**b**) indicates the fungal community.

**Table 3.** RDA and soil and cover grass properties correlated with the soil microbial communities at the genus level.

| | Bacteria | | | | Fungi | | | |
|---|---|---|---|---|---|---|---|---|
| | RDA1 | RDA2 | $r^2$ | $p$ | RDA1 | RDA2 | $r^2$ | $p$ |
| T | 0.991 | 0.128 | 0.113 | 0.282 | 0.083 | −0.996 | 0.105 | 0.272 |
| SWC | −0.991 | −0.128 | 0.036 | 0.648 | 0.309 | −0.951 | 0.004 | 0.950 |
| pH | 0.918 | −0.395 | 0.623 | 0.001 *** | 0.540 | −0.841 | 0.077 | 0.407 |
| AN | −0.670 | 0.742 | 0.347 | 0.007 ** | −0.238 | 0.971 | 0.034 | 0.657 |
| TN | −0.872 | 0.488 | 0.366 | 0.007 ** | 0.363 | 0.931 | 0.096 | 0.305 |
| TP | −0.452 | 0.891 | 0.052 | 0.577 | −0.275 | 0.961 | 0.285 | 0.019 * |
| TK | −0.710 | 0.703 | 0.203 | 0.069 | 0.150 | 0.988 | 0.337 | 0.010 ** |
| AP | −0.984 | 0.177 | 0.279 | 0.029 * | −0.172 | 0.985 | 0.452 | 0.005 ** |
| AK | −0.946 | 0.322 | 0.372 | 0.011 * | −0.421 | 0.906 | 0.250 | 0.047 * |
| SOC | −0.905 | 0.423 | 0.262 | 0.036 * | 0.157 | 0.987 | 0.237 | 0.046 * |
| MBC | 0.674 | 0.737 | 0.062 | 0.501 | 0.519 | −0.854 | 0.108 | 0.273 |
| MBN | 0.821 | 0.569 | 0.085 | 0.355 | 0.980 | −0.196 | 0.395 | 0.015 * |
| GB | 0.784 | −0.620 | 0.092 | 0.337 | −0.996 | −0.087 | 0.102 | 0.294 |
| GC | 0.783 | −0.620 | 0.285 | 0.034 * | 0.246 | −0.969 | 0.104 | 0.303 |
| GN | 0.855 | −0.517 | 0.242 | 0.053 | 0.589 | −0.807 | 0.062 | 0.506 |
| GP | 0.762 | −0.647 | 0.317 | 0.018 * | 0.681 | −0.731 | 0.037 | 0.643 |

T indicates soil temperature, SWC indicates soil water content, SOC indicates soil organic carbon, AN indicates available nitrogen, TN indicates total nitrogen, TP indicates total phosphorus, AP indicates available phosphorus, AK indicates available potassium, TK indicates total potassium, MBC indicates microbial biomass carbon, MBC indicates microbial biomass nitrogen, GB indicates cover grass biomass, GC indicates cover grass carbon, GN indicates cover grass nitrogen and GP indicates cover grass phosphorus. *** ($p < 0.001$), ** ($p < 0.01$) and * ($p < 0.05$). The $r^2$ indicates the proportion of variance explained.

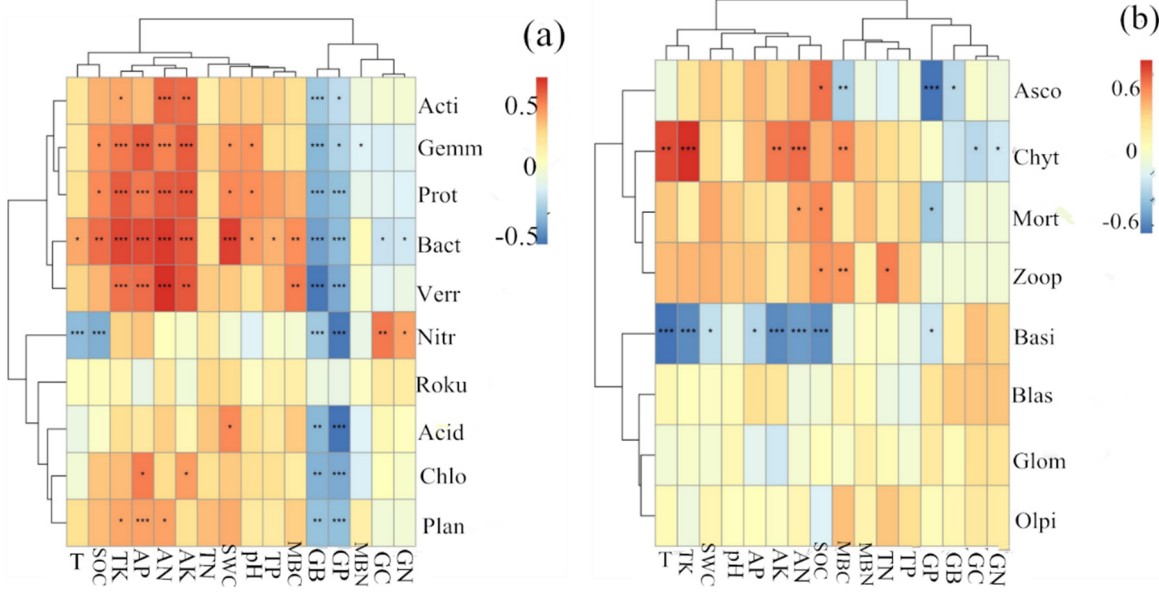

**Figure 4.** Spearman's rank correlation coefficients describing the relationships between the soil properties, cover grass properties and microbial community compositions. T indicates temperature, SWC indicates soil water content, SOC indicates soil organic carbon, AN indicates available nitrogen, TN indicates total nitrogen, TP indicates total phosphorus, AP indicates available phosphorus, AK indicates available potassium, TK indicates total potassium, MBC indicates microbial biomass carbon, MBC indicates microbial biomass nitrogen, GB indicates cover grass biomass, GC indicates cover grass carbon, GN indicates cover grass nitrogen, and GP indicates cover grass phosphorus. (**a**) indicates the bacterial community and (**b**) indicates the fungal community. Prot indicates Proteobacteria, Acid indicates Acidobacteria, Acti indicates Actinobacteria, Chlo indicates Chloroflexi, Gemm indicates Gemmatimonadetes, Bact indicates Bacteroidetes, Plan indicates Planctomycetes, Roku indicates Rokubacteria, Nitr indicates Nitrospirae, Verr indicates Verrucomicrobia. Asco indicates Ascomycota, Chyt indicates Chytridiomycota, Mort indicates Mortierellomycota, Basi indicates Basidiomycota, Zoop indicates Zoopagomycota, Blas indicates Blastrociadiomycota, Glom indicates Glomeromycota, Olpi indicates Olpidiomycota. *** ($p < 0.001$), ** ($p < 0.01$) and * ($p < 0.05$).

## 4. Discussion

### 4.1. Changes in Soil Characteristics after Cover Grass Treatment over the Short Term

Cover grass reduced the soil total N and available N in the apple orchard, but these levels were not significantly affected by cover grass intensity in the short term. These results are similar to those of early studies, which showed that cover grass reduced soil $NO_3^-$-N content in a vineyard [37]. In contrast, some previous studies have shown that cover crops or grass can increase total soil N [8,38,39]. These opposite conclusions may be related to the cover grass treatment duration. In the early stage of cover grass treatment in the orchard, there was competition between the apple trees and the grass for nutrients in the soil; however, over the duration of the cover grass treatment, organic residues, litter and root secretions of the cover grass entered the soil, thereby improving the soil environment and soil nutrients [40,41].

### 4.2. Soil Microbial Community during Cover Grass Treatment

The dominant aboveground vegetation can largely determine the composition of the soil microbial community [1,42]. In the apple orchard, the greatest determinant of the composition of the soil microbial community was the apple trees [43], a result that was consistent with Zheng et al. [8]. Thus, soil microbial community diversity was not obviously changed by cover grass in the apple orchard, which are similar to the findings of Wei et al. [19] who reported that cover cropping had no significant effects on soil bacterial diversity in mango orchards, this is because of the primary aboveground vegetation and litter composition did not change fundamentally because of the cover grass treatment over a short-term period.

Consistent with our hypothesis, the primary groups in the microbial communities changed after cover grass was planted, and variations in the dominant soil bacterial phyla were primarily caused by differences in Proteobacteria, Chloroflexi, Gemmatimonadetes, Bacteroidetes and Nitrospirae among the different treatments. Previous studies have reported that the relative abundance of superior soil bacterial phyla, such as Proteobacteria, Firmicutes, Acidobacteria, Bacteroidetes and Actinobacteria, usually changed after cover crop treatment [44,45]. Proteobacteria and Bacteroidetes are generally considered copiotrophic bacteria [46–48] and Chloroflexi is one of the bacterial phyla that can survive in soil under stressful conditions with poor organic matter [49,50]. Although cover grass had no significant effects on soil bacterial diversity in our study, it significantly influenced soil bacterial community compositions, which is also similar to the findings by Wei et al. [19]. Our study results showed that Proteobacteria and Bacteroidetes abundances decreased in the low-intensity cover grass, while Chloroflexi abundance increased in low-intensity cover grass (20%), which together indicated that the soil fertility of the apple orchard under the low-intensity cover grass treatment became more highly oligotrophic; we also found that Bacteroidetes abundance decreased with moderate-intensity (40% and 60%) cover grass, showing that the apple orchard with the moderate-intensity cover grass treatment was oligotrophic to some extent. In combination with the finding that cover grass reduced the soil nitrogen content, we can infer that the soil nutrient conditions of the apple orchards with low-intensity and moderate-intensity cover grass treatments were reflected by the dominant bacterial phyla. Gemmatimonadetes adapts to low soil moisture and higher Gemmatimonadetes abundances occur in soils at a closely neutral pH than in acidic soils [51–54]. Gemmatimonadetes abundance also decreased with moderate-intensity (60%) and high-intensity (80%) cover grass, suggesting that the apple orchard with a higher intensity of cover grass had higher soil moisture. Nitrospirae can participate in soil nitrogen cycling [55]. The relative abundance of Nitrospirae also increased under high-intensity cover grass, indicating that high-intensity cover grass altered nitrogen cycling in the apple orchard soil. However, the soil nutrient condition of the apple orchard with high-intensity cover grass was not clearly reflected by the dominant bacterial phyla.

The changes in dominant soil fungal phyla were driven by Ascomycota and Mortierellomycota between the different measures (Figure 1 and Table S2), which are similar to

Castle et al. [56] who reported that after three years of cropping, fungal communities respond to cover crops. Unfortunately, the dominant fungal phyla were not apparently different between the NC and the other grass cover measures. However, we observed that cover grass altered the apple orchard soil fungal community composition (Figure 2b). Based on these findings, we suspect that changes in some fungal groups in the soil caused by cover grass are not reflected at the phylum level.

*4.3. Effects of Soil Properties and Cover Grass on Microbial Communities*

Gemmatimonadetes was remarkably positively related to SOC, AN and AP, which was consistent with Liu et al. [57], who revealed that Gemmatimonadetes was obviously responsive to the C, N and P standards. Actinobacteria abundance was positively correlated with AN, which was consistent with Ramirez et al. [58], who found that Actinobacteria abundance increased with increasing nitrogen content. In addition, we also found that Actinobacteria, Proteobacteria, Bacteroidetes, Gemmatimonadetes, Chloroflexi and Verrucomicrobia were remarkably positively related to TK and AK, which showed that these six bacterial phyla were sensitive to the soil K level. The RDA results showed that SOC, AN, AP and AK obviously affected the bacterial communities (Figure 3 and Table 3), which further confirmed that soil C, N, P and K play a key role in building microbial communities. Simultaneously, the C and P contents of the cover grass also obviously affected the bacterial communities. This result may be because carbon that aboveground plants put into the soil (via root exudates) is the primary carbon source used by the bacterial community [59,60]; additionally, phosphate-solubilizing bacteria influence the phosphorus absorption of aboveground vegetation [61].

Our results showed that TP, TK, AP, AK, SOC and MBN were remarkably related to the fungal communities (Figure 3 and Table 3). Studies have shown that Ascomycota was positively related to AP [62,63]; hence, phosphorus was an important factor that correlated with fungal communities [63,64]. The abundances of Bacteroidetes and Chytridiomycota were correlated with TK and AK; combined with the RDA results, these findings further confirm that K was also an important factor related to fungal communities in the orchard.

## 5. Conclusions

In summary, cover grass had no impact on microbial community diversity, but it significantly changed the microbial community composition. Cover grass altered the bacterial community; the soil copiotrophic groups (e.g., Proteobacteria) had the lowest abundance in the low-intensity (20%) cover grass. The mild cover grass treatments (20% and 40%) altered the apple orchard soil fungal community composition in the short term; fungal community composition did not exhibit significant differences at the phylum level in the different cover grass intensity treatments relative to the no cover grass treatment. Moreover, the changes in the soil microbial community at the given phyla may be related to the change in soil AN content caused by cover grass; the cover grass properties influenced the soil bacterial communities rather than the fungal communities.

**Supplementary Materials:** The following are available online at https://www.mdpi.com/article/10.3390/f12121787/s1, Figure S1: Rarefaction curves of the bacterial (a) and fungal (b), Figure S2: Relative abundance of the dominant groups of the bacterial (a) at the order level and fungal (b) communities at the class level among the different treatments. A indicates 0% (CK), B indicates 20% (LIC), C indicates 40% (MIC1), D indicates 60% (MIC2), and E indicates 80% (HIC). Table S1: Quality assessment of bacterial sequencing data, Table S2: Quality assessment of fungal sequencing data, Table S3: Composition of bacterial community at the phylum level, Table S4: Composition of bacterial community at the orders level, Table S5: Composition of fungi community at the phylum level, Table S6: Compositions of fungi community at the classes level.

**Author Contributions:** P.W. designed the experiments and collected data; P.W. and R.H. analyzed the data; P.W. wrote the manuscript. All authors discussed the results and reviewed the manuscript. All authors have read and agreed to the published version of the manuscript.

**Funding:** This work was financially supported by the financial support of the National Natural Science Foundation of China (32001311) and the Fundamental Research Funds for the Central Universities (2452021017).

**Institutional Review Board Statement:** Not applicable.

**Informed Consent Statement:** Not applicable.

**Data Availability Statement:** The data and results of this study are available upon reasonable request. Please contact the main author of this publication.

**Conflicts of Interest:** The authors have declared no competing interests exist.

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
