# Peer review of "Short-Term Effects of Cover Grass on Soil Microbial Communities in an Apple Orchard on the Loess Plateau"

_forests, doi:10.3390/f12121787_

Round 1

Reviewer 1 Report

The evaluated paper is interesting and in general is well prepared. English quality is good. Authors should pay more attention on interpunction and missing spaces, of which there are many in the text. Also References style at the end of ms is unacceptable in the current form and must be corrected according Forest rules. I also suggest some changes in the text in order to improve ms quality:

Line 11: I suggest to change abbreviation of "no cover" as NC, instead of KC. Please correct it in the whole ms text

Line 21: Abbreviation of AN should be explained in the place of first using

Line 30: should be [1-3]

Line 34: should be [6-8] and also I suggest to add here the following citation: WoliÅ„ska A., Górniak D., Zielenkiewicz U., Goryluk-Salmonowicz A., Kuźniar A., StÄ™pniewska Z., BÅ‚aszczyk M., 2017. Microbial biodiversity in arable soils is affected by agricultural practices. International Agrophysics, 31(2), 259-271,

Line 36: should be [9-11]

Line 38: should be [11, 14-16]

Line 47: should be [22-24]

Line 107: please add an information about weight of soil used for DNA extraction and specify if the isolation was in 1 repetition or in 3?

Line 173: At the beginning of NGS analysis the general information about total number of sequences should be noted. I suggest to add the Table with a summary of the sequencing data quality obtained in the current study. In this Table the following data should be presented: input – number of reads in raw fastq files; filtered – number of reads after preliminary quality filtering; denoised (F/R) – number of reads after quality filtering; merged – number of merged forward-reverse reads; nonchim – number of merged reads after removal of chimera sequences; % passed – relative number of passed reads after all the above steps.

Also rarefaction curve should be included in ms or in Supplementary material.

Line 229: Quality of Figure 3 must be improved as taxonomic levels are unreadable

Line 354: References section demands to be rewritten and corrected according Forest rules, the current style of citations  is improper.

Author Response

Dear editor and reviewers:

On behalf of my co-authors, I would like to write this letter concerning your comments on our manuscript “Short-term effects of cover grass on soil microbial communities in an apple orchard on the Loess Plateau” (Submission no: forests-1490676). After reading the reviewers comments carefully, we revised the manuscript following your helpful advices. The responses are listed below. We highly appreciate your comments and efforts to help us improve the paper.

Kindest regards,

Pan Wan

Reviewer

Responses:

-- 1. Line 11: I suggest to change abbreviation of "no cover" as NC, instead of KC. Please correct it in the whole ms text.  

Response: Thanks for your careful reading of our manuscript. We have changed abbreviation of "no cover" as NC in our manuscript.

-- 2. Line 21: Abbreviation of AN should be explained in the place of first using

Response: I am sorry for these mistakes. Following your comments, we modified these mistakes as you suggested.

--3. Line 30: should be [1-3]

Response: Thanks for your detailed comments, we modified these mistakes as you suggested.

--4. Line 34: should be [6-8] and also I suggest to add here the following citation: WoliÅ„ska A., Górniak D., Zielenkiewicz U., Goryluk-Salmonowicz A., Kuźniar A., StÄ™pniewska Z., BÅ‚aszczyk M., 2017. Microbial biodiversity in arable soils is affected by agricultural practices. International Agrophysics, 31(2), 259-271.

Response: Thanks for your detailed comments, we modified these mistakes as you suggested. Meanwhile, we have add here the following citation: WoliÅ„ska A., Górniak D., Zielenkiewicz U., Goryluk-Salmonowicz A., Kuźniar A., StÄ™pniewska Z., BÅ‚aszczyk M., 2017. Microbial biodiversity in arable soils is affected by agricultural practices. International Agrophysics, 31(2), 259-271.

--5. Line 36: should be [9-11], Line 38: should be [11, 14-16],Line 47: should be [22-24]

Response: We modified these mistakes as you suggested.

--6. Line 107: please add an information about weight of soil used for DNA extraction and specify if the isolation was in 1 repetition or in 3?

Response: I am sorry for these mistakes, for your comments, we have added the information about weight of soil used for DNA extraction. In this study, 25 plots were established, each cover grass treatment was 5 replicates. Hence, 25 composite samples were collected, namely, each treatment was 5 replicates soil samples. 5 replicates for each analysis in our study. 0.25 g of each soil sample used for DNA extraction from each soil sample (25 soil samples). 5 replicates for each analysis in our study.

--7. Line 173: At the beginning of NGS analysis the general information about total number of sequences should be noted. I suggest to add the Table with a summary of the sequencing data quality obtained in the current study. In this Table the following data should be presented: input – number of reads in raw fastq files; filtered – number of reads after preliminary quality filtering; denoised (F/R) – number of reads after quality filtering; merged – number of merged forward-reverse reads; nonchim – number of merged reads after removal of chimera sequences; % passed – relative number of passed reads after all the above steps. Also rarefaction curve should be included in ms or in Supplementary material.

Response: Thanks for your comments, we have added the Table S1and S2 with a summary of the sequencing data quality obtained in the Supplementary material. In addition, following your comments, we have added the rarefaction curve in Supplementary material. We hope that you agree.

--8. Line 229: Quality of Figure 3 must be improved as taxonomic levels are unreadable

Response: Following your comments, we have reproduced figures in our manuscript. We hope that you agree.

--9. Line 354: References section demands to be rewritten and corrected according Forest rules, the current style of citations is improper.

Response: Thanks for your comments, we modified the references section in our manuscript according Forests rules.

Finally, we greatly appreciate your comments and advises that helped us to improve the manuscript. Thank you very much for your time and effort.

Reviewer 2 Report

This manuscript analysed soil properties, microbial community diversity and composition in an apple orchard after being covered with native wild grasses for 3 years on the Loess Plateau, China. The ratios of cover grass were 0% (no cover, CK), 20% (low-intensity cover, LIC), 40% (moderate-intensity cover, MIC1), 60% (moderate-intensity cover, MIC2) and 80% (high-intensity cover, HIC). Meanwhile, the relationships between soil nutrients, cover grass properties, and microbial communities was analysed by Redundancy analysis and Pearson correlations. The results showed that cover grass altered the bacterial community composition, and significant changes at the phylum level were mainly caused by Proteobacteria, Bacteroidetes, and Chloroflexi. Compared with CK, the abundance of Proteobacteria was lower in LIC, and the abundance of Bacteroidetes was lower in LIC, MIC1 and MIC2, while that of Chloroflexi was higher in LIC. LIC and MIC1 were the only cover grass intensities that altered the soil fungal community composition; there were no significant differences at the phylum level. 

Therefore, this study has been well conducted and provided important results. However, some improvements and revisions are still required as shown below;

- In the introduction section:

Please include more recent literatures highlighting the recent background on this topic.

Also, write the full objective of this work at the last paragraph of Introduction section.

- In Methods:

Please state the number of replicates you performed for each analysis, and cite all the methods you used.

In Results:

All figures are of low resolution and can not be read well. Please replace them with high quality figures.

Table 2 needs more explanation to highlight its important results.

- In Discussion:

The discussion should be improved and interpreted with the results and compared with the previous published related work

- English of the whole manuscript text should be revised and proofread by English native professional.

Author Response

Dear editor and reviewers:

On behalf of my co-authors, I would like to write this letter concerning your comments on our manuscript “Short-term effects of cover grass on soil microbial communities in an apple orchard on the Loess Plateau” (Submission no: forests-1490676). After reading the reviewers comments carefully, we revised the manuscript following your helpful advices. The responses are listed below. We highly appreciate your comments and efforts to help us improve the paper.

Kindest regards,

Pan Wan

Reviewer

Responses:

-- 1.- In the introduction section: Please include more recent literatures highlighting the recent background on this topic. Also, write the full objective of this work at the last paragraph of Introduction section.

Response: Thanks for your careful reading of our manuscript. Following your comments, we have added the recent literatures highlighting the recent background on this topic. In addition, we have added the full objective of this work at the last paragraph of Introduction section. We hope that you agree.

-- 2. - In Methods: Please state the number of replicates you performed for each analysis, and cite all the methods you used.

Response: I am sorry for these mistakes. In our study, 25 plots were established, each cover grass treatment was 5 replicates. Hence, 25 composite samples were collected, namely, each treatment was 5 replicates soil samples, and 5 replicates for each analysis. Following your comments, we have revised the section of Methods and provided more detailed in our Method section. In addition, we have cited all the methods our used. We hope that you agree.

-- 3 In Results: All figures are of low resolution and can not be read well. Please replace them with high quality figures.Table 2 needs more explanation to highlight its important results.

Response: I am sorry for these mistakes. Following your comments, we have reproduced figures in our manuscript. In addition, we have added the Table S1and S2 with a summary of the sequencing data quality obtained in the Supplementary material. We hope that you agree.

-- 4 In Discussion: The discussion should be improved and interpreted with the results and compared with the previous published related work

Response: Following your comments, we have revised the section of Discussion and compared with the previous published related work. We hope that you agree.

-- 5 English of the whole manuscript text should be revised and proofread by English native professional.

Response: The English in this document has been checked by MDPI Editing Services, and we read the manuscript carefully after the English revision to convince that the text was revised correctly.

For a certificate, please see the chart below:

Finally, we greatly appreciate your comments and advises that helped us to improve the manuscript. Thank you very much for your time and effort.

Round 2

Reviewer 2 Report

The revised manuscript is greatly improved as per my suggested comments 

Author Response

We greatly appreciate your comments that helped us to improve the manuscript.